# Quantum Transport in Large-Scale Patterned Nitrogen-Doped Graphene

**DOI:** 10.3390/nano13182556

**Published:** 2023-09-14

**Authors:** Aleksander Bach Lorentzen, Mehdi Bouatou, Cyril Chacon, Yannick J. Dappe, Jérôme Lagoute, Mads Brandbyge

**Affiliations:** 1Department of Physics, Technical University of Denmark, DK-2800 Kongens Lyngby, Denmark; abalo@dtu.dk; 2Laboratoire Matériaux et Phénomènes Quantiques, CNRS-Université Paris Cité, 10 Rue Alice Domon et Léonie Duquet, CEDEX 13, 75205 Paris, France; mehdi.bouatou@univ-paris-diderot.fr (M.B.); cyril.chacon@univ-paris-diderot.fr (C.C.); jerome.lagoute@univ-paris-diderot.fr (J.L.); 3SPEC, CEA, CNRS, Université Paris-Saclay, CEA Saclay, CEDEX, 91191 Gif-sur-Yvette, France; yannick.dappe@cea.fr

**Keywords:** graphene, nitrogen doping, patterning, quantum transport calculations, tight-binding model

## Abstract

It has recently been demonstrated how the nitrogen dopant concentration in graphene can be controlled spatially on the nano-meter scale using a molecular mask. This technique may be used to create ballistic electron optics-like structures of high/low doping regions; for example, to focus electron beams, harnessing the quantum wave nature of the electronic propagation. Here, we employ large-scale Greens function transport calculations based on a tight-binding approach. We first benchmark different tight-binding models of nitrogen in graphene with parameters based on density functional theory (DFT) and the virtual crystal approximation (VCA). Then, we study theoretically how the random distribution within the masked regions and the discreteness of the nitrogen scattering centers impact the transport behavior of sharp n−p and $n-n^{\prime }$ interfaces formed by different, realistic nitrogen concentrations. We investigate how constrictions for the current can be realized by patterned high/low doping regions with experimentally feasible nitrogen concentrations. The constrictions can guide the electronic current, while the quantized conductance is significantly washed out due to the nitrogen scattering. The implications for device design is that a p−n junction with nitrogen corrugation should still be viable for current focusing. Furthermore, a guiding channel with less nitrogen in the conducting canal preserves more features of quantized conductance and, therefore, its low-noise regime.

## 1. Introduction

The electronic transport properties of graphene and 2D nanostructures based on graphene are truly remarkable. Graphene and graphene nanoribbons are quantum conductors which feature only a few electronic states/modes for transport, but with very low intrinsic electron–phonon scattering, related to the inherent symmetry properties of the quantum states in these materials. The path electrons travel on average before scattering can go beyond micro-meters, even at room temperature, and quantum wave interference effects can survive up to room temperature in graphene nanoribbons [1,2]. These properties have promoted ideas of using electrons in graphene, behaving as Dirac fermions, for two-dimensional electron optics where the electrons behave like photons, or devices based on Klein tunneling [3]. This, however, requires well-defined and sharp interfaces at the Fermi wave-length scale ($\lambda_{F}=2\pi /k_{F}$) between regions of different carrier density in the form of p−n (holes are carriers on one side, electrons the other) or $n-n^{\prime }$ (different electron concentrations) [4] junctions, to enable electron current focusing or collimation, and realize various components such as Veselago lenses [5,6], gratings [7], or novel transistor devices [8]. For example, to achieve effective focusing without reflecting a substantial amount of the current requires junction widths below 10 nano-meters [5], which is difficult to achieve using electrostatic gating. Moreover, it requires charges at atomic-scale distances to the graphene plane, either by charge transfer from a nearby van der Waals material [9] or through in-plane doping.

Nitrogen-doped graphene has been focused on as a wide interest for fundamental and applied research in various fields, as outlined by several review articles [10,11,12,13]. For in-plane doping, it is a natural choice to dope graphene by substituting carbon with nitrogen, due to similar atomic radii and sp${}^{2}$-bonding with similar bond-lengths, leading to low lattice deformation and planarity. This, furthermore, gives rise to a π-system of electrons at the Fermi-level, which is well-described by a widely used $p_{z}$-model. Recently, it was shown how it may be possible to introduce patterns of N-doped regions on graphene by using a molecular overlayer mask of $C_{60}$-molecules in the $N^{+}$-ion implantation, enabling a spatial doping control down to the nano-meter scale [14]. In Figure 1A, we show Scanning Tunneling Microscopy (STM) images of a $C_{60}$ overlayer mask on graphene. Here, the outermost graphene layer of a slab of highly oriented pyrolytic graphite (HOPG) is seen.

When doping graphene in the lab, the different concentrations of nitrogen are implanted by first covering parts of the graphene with a C${}_{60}$ mask, then bombarding the surface with nitrogen, yielding a 60% reduction in nitrogen implantation under the mask [14]. This gives a potential profile that corresponds to the shape of the covering layer when the C${}_{60}$ is removed from the surface by sweeping the STM tip. Other interesting methods are also available for doping graphene, such as the annealing of graphene oxide [15], but lack the nano-scale precision of the method developed in ref. [14].

Figure 1B illustrates how the nitrogen concentration is controlled by the mask, and how the mask can be removed using the STM tip after exposure (Figure 1), creating interfaces between different doping concentrations. The typical doping levels are from 0.01–0.2% nitrogen substitution [14].

The doping nitrogen will induce carrier scattering, since the nitrogen atom is acting as a screened, artificial proton potential [16]. This scattering potential will, however, depend on the surrounding carrier concentration dictated by the number of nearby nitrogens, and possibly additional carriers controlled by an external gate potential. The method based on the masked ion-implementation will introduce some randomness in the dopant positions and, while the overall doping concentrations may be controlled, it is clear that the interfaces between regions will have inherent disorder (cf. Figure 1B). Thus, it is interesting to assess to what degree we may expect this scattering and disorder to ruin the electron optics properties of ideal p−n and $n-n^{\prime }$ junctions, conductance quantization, and shot-noise transport properties of constrictions formed between regions of different doping.

In this work, we employ transport calculations based on Greens functions of the interface between high and low N-doping regions in graphene. We consider parameters which are comparable to the actual experiments by Bouatou et al. [14] (Figure 1). We first compare different tight-binding parametrizations, and benchmark transport obtained with tight-binding to the corresponding results based on density functional theory (DFT) for small systems. Then, we consider transport in large-scale tight-binding transport calculations of the electron current flow and transmissions in planar junctions and constrictions, to assess the role of N-scattering/disorder on electron beam focusing and conductance quantization, respectively.

## 2. Methods and Parameters

In order to describe the doping concentration corresponding to N–N distances on the order of 10nm, we need device regions involving beyond $10^{5}$ atoms. This renders pure first principle approaches at the DFT level impossible. Instead, we use a tight-binding approach where we compare tight-binding to DFT calculations regarding different aspects of the junctions. In particular, we will consider the total system consisting of a delocalized doping charge, defined in the different regions together with a more local perturbation around each N dopant.

We first consider the screening and junction width from a DFT calculation of an ideal, straight junction between two regions corresponding to a jellium-like description where the doping is simulated using the virtual crystal approximation(VCA) [17]. In VCA, the charge donated from the nitrogen atoms is distributed in the core of each carbon atom by including it in the charge of the pseudopotentials on all carbon atoms on each side of the junction, and, thus, is smeared out as an approximation. The DFT calculations here are conducted using TranSiesta [18] with a single-zeta polarized basis set, enabling us to extract the potential shift in the $p_{z}$ orbitals across the junction. We used the Perdew–Burke–Ernzerhof (PBE) exchange–correlation functional together with 20 *k*-points in the transverse (*y*-) direction and an electron smearing of 0.025 eV.

The results are shown in Figure 2. The carrier charge, including doping, is then screened at the interface between the different regions. The junction width varies slightly but is on the order of 5nm, in agreement with the experiments in [14]. The 0.2%/0.0% junction notably has a longer decaying tail on the side with no charge doping compared to the other cases. We note that, generally, the amount of conduction electrons in the device can be modified in experiments with a variable back gate potential, which then should also give rise to a slightly variable junction width as the gating potential is changed, according to Figure 2.

### 2.1. Tight-Binding Models

For pristine graphene, we employ a non-orthogonal tight-binding model [19,20], where the carbon atom is modeled using a single $p_{z}$-orbital (hopping-element of t=−2.7 eV and overlap of s=0.11). We split the contribution of the nitrogen dopants into a smooth background potential, as modeled in the spirit of VCA, corresponding to the spatial variation in the Dirac point measured by STM spectroscopy [14], and a short-ranged scattering potential centered at the nitrogen atoms. This is schematically shown in Figure 3A. Nitrogen atoms are described by a single $p_{z}$-orbital, and the dominant effects of the individual N are included through position-dependent on-site shift, $V_{N}\left(r\right)$, to the surrounding carbon atoms. The modification of the couplings coming from the added potential is neglected, as only energies close to the Fermi-level are considered. A number of different model choices for $V_{N}\left(r\right)$ have been considered previously [19,21,22]. Here, we employ the parameters given by Lambin et al. [19], where the nitrogen on-site potential distance dependence was fitted to a single Gaussian (Model 1 in Figure 3B) or a sum of two Gaussians (Model 2 in Figure 3B), which were obtained by comparing to a DFT calculation on nitrogen substitutions in long (10, 10) carbon nanotubes [21]. The latter model fit our DFT calculations well for planar systems, as illustrated in Figure 3B.

The potential landscape from Figure 3A will be a built-in potential in the graphene sheet, meaning the Dirac point on either side of the junction will stay fixed (and different). The Fermi-level can be tuned by a backgate [23], meaning that we could expect a situation where the $n-n^{\prime }$ junction could be adjusted with an appropriate backgate voltage to become a p−n junction.

Furthermore, we are here treating the N atoms as isolated and which yield n-doping, in agreement with the experiments. However, we note that, for N-atoms at much closer distances, the electronic structure may change significantly leading to p-doping instead. This has been seen in doping using solution plasma, leading to a new material (cationic nitrogen-doped graphene, CNG), with 13.4% nitrogen [24].

### 2.2. Benchmark Transport Calculations

Next, we compare transport calculations based on the $p_{z}$-only tight-binding (TB) model in Figure 3 to full DFT, for a planar junction plus a nitrogen atom at different positions with respect to the junction. Here, we employ periodic boundary conditions in the transverse-to-transport direction using a *k*-point sampling, $n_{k}=400$, for electronic transmission. In Figure 4, we show the transmission function around the Fermi energy ($E_{F}$), calculated using the TranSiesta/TBtrans [25] and SISL [26] tools. Here, we see that the TB fits the full DFT reasonably well for the different N positions in the occupied part of the spectrum, while it misses resonances when $E-E_{F}>0.2\;\mathrm{eV}$. However, as long as we restrict our considerations to energies below these resonances, the $p_{z}$ model should describe the system well.

### 2.3. Large-Scale Transport

The large-scale tight-binding transport calculations [20,26,27] are performed using a simulation region consisting of the orthogonal graphene unit-cell tiled out to a large cell $N_{x}\times N_{y}$ in vertical/horizontal directions using the SISL tool [26]. We then change the on-site potential corresponding to the doping profile and distribution of nitrogen dopants. In order to characterize the behavior of the current flow in the system, we calculate the spectral bondcurrent as implemented in TBtrans, and plot it using the technique and methods described in Refs. [20,25]. In particular, complex absorbing potentials (CAPs) are used to absorb current and avoid spurious edge effects which would appear in a (finite) ribbon approximation [20,28].

## 3. Results

### 3.1. Straight p−n Junction without and with Nitrogen Dopants

We first consider how a straight/uncorrugated n−p junction with a width and doping can focus or collimate an electron current injected from a point source such as an STM-tip [6]. In our calculation, the current is being injected from a carbon site placed about 10 nm from the junction at the center of the star pattern seen in Figure 5A. This point injection is modeled using a wideband electrode coupling $\Gamma_{\mathrm{tip}}\approx 0.7$ eV. We imagine that we may tune the energy (tip voltage) of the injected electrons to maximize the degree of focusing and choose it such as in the following.

In Figure 5, we consider the electron focusing properties of the p−n-junction with a width comparable to those fabricated using the mask-technique in Figure 1 and compare the situations without Figure 5A and including the nitrogen scattering centers Figure 5B–D. In Figure 5B, we employ a TB parametrization based on DFT T-matrix calculations [22], where the scattering from N is just described by the onsite potential at the N-site. Although this potential is considerably stronger than the Lambin [19] model, it corresponds to a scattering similar to a 50% reduction in the $V_{N}$ by Lambin as shown qualitatively in Figure 5C. This illustrates how the distance dependence in the nitrogen potential plays an important role. However, we note that effects not accounted for in these models, such as screening by the substrate or delocalization errors intrinsic to DFT, might reduce $V_{N}^{L}$ further. In all cases, for these rather different models, we find that it is possible to find energy ranges, *E*, where the current pattern displays focusing. The results indicate that a gating-potential in a device may be used to tune the Fermi energy to obtain current focusing from point injection, despite the disorder introduced by the random dopants.

### 3.2. Constrictions from p−n Junctions in Graphene

When doping graphene in the lab, the different concentrations of nitrogen are implanted by first covering parts of the graphene with C${}_{60}$ molecules, then bombarding the surface with nitrogen [14]. This gives a potential profile that corresponds to the shape of the cover layer/mask where the covered parts then obtain a lower nitrogen doping compared to the uncovered parts. In Figure 6A, an experimental STM image is shown displaying a number of constrictions formed in the C${}_{60}$ covering layers. Therefore, it is interesting to consider, theoretically, the transport properties of one such constriction and to what extent the patterning may be used to guide electronic current.

There has been considerable interest in fabricating constrictions in graphene nanostructures to show ballistic transport and conductance quantization phenomena [29,30,31,32]. The conductance quantization and demonstration of wave propagation in separate, well-defined modes has been observed for graphene nanoribbons with movable probes [33]. However, this has turned out to be more challenging for fabricated constrictions due to the role of edge disorder and edge channels [31]. Ideally, the transmission function should increase in discrete steps of $G_{0}=2e^{2}/h$ for spin-degenerate modes, and with another factor of 2 if valley degeneracy is present (i.e., for non-edge modes) in a slowly varying potential [34].

In Figure 6B, we show our theoretical constriction model consisting of two regions and associated potentials, with a gradual transition between them. The width of the interface between the two potential regions is kept fixed at roughly 5 nm. Thus, we neglect the changes in the width with a variable amount of screening charge, cf. Figure 2. We denote the case with a high doped constriction ($n^{\prime }$ doped region 1, $E_{D1}=-0.24$ eV) surrounded by a low doped region (*n* doped region 2, $E_{D2}=-0.14$ eV) as the "normal" case corresponding to the masked region 2, cf. Figure 6, while we use the term “negative” for the reversed case. We apply semi-infinite electrodes (left/right in Figure 6B) and CAPs in the transverse directions (top/bottom).

In Figure 7, we show the transmission function where very smoothened step-like features at $G=2G_{0}\times n$ are seen when no nitrogen scatterers are included. For the “normal”/“negative” case, the steps are only present above/below the charge neutrality point of the constriction part ($V_{1}$ in Figure 6), indicating an asymmetry in the conduction of electrons and holes which is close to mirror-opposite for each type of system. The steps are down-shifted from the even-*n* levels, as one could expect from a leaky waveguide where parts of the injected current (from the Left) enters the barrier region and is absorbed by the CAP. However, in both “negative”/“normal” cases there is a significant additional scattering and, thus, lower transmission when the scattering potentials of nitrogen are introduced. Here, it is only the “negative” case which displays step features in the transmission function, but at about 50% reduced transmission compared to the pristine case. This comes down to the “negative” constriction having less nitrogen scatterers inside and around the constriction.

This potential scattering is also clearly visible in the corresponding bondcurrents shown in Figure 8C. For the “normal” case, as in Figure 6A, a clear current guiding effect is observed in Figure 8A,B in the bondcurrents, demonstrating how the current preferentially flows in the low doped *n*-channel between the wider regions.

In Figure 9 and Figure 10 we compare the transmissions as a function of energy and constriction width for the “negative” and “normal” cases, respectively. We contrast three cases of regions where the N scattering potential is included: nowhere (clean), outside, and in all of the constriction region. The steps from the clean case in Figure 7 are also found in Figure 9A and Figure 10A, and partly in Figure 9B,C (negative case with nitrogen). In the “negative” case the step-like structure of the transmission function is significantly affected, while it is completely washed out in the “normal” case. Especially, for the narrowest “negative” constriction with W=15 nm a more clear step feature can be seen near E=−0.25 eV, which is washed away when the constriction gets wider. This might reflect the higher energy spacing between transverse modes proportional to 1/W in the narrowest constriction giving less inter-mode scattering [35]. More comments on this is given in the discussion.

In order to analyze the step-like transmission functions, T(E), and see if they correspond to well-defined modes, we examine the eigenvalues, $\tau_{n}$ where $T=\sum_{n}\tau_{n}$, of the transmission probability matrix of the different systems [36], in Figure 11 (“negative” case) and Figure 12 (“normal” case).

Without any nitrogen scatterers, as shown in Figure 9A and Figure 10A, the transmission eigenvalues are, to some extent, appearing in separate integer steps, albeit with a small shoulder before approaching unity. In this clean case there is almost perfect valley degeneracy in the eigenchannel transmission, only broken due to the smooth potential in Figure 9A and Figure 10A, which also break this symmetry slightly and it is almost unnoticeable.

When nitrogen scattering potentials are introduced in the device, only the “negative” case in Figure 11B,C has step features resembling the clean constriction case with some distortion. Even here, stronger variation with energy is seen in the eigenchannel transmissions, and the “shoulder” in the first transmission eigenvalue becomes more noticeable, especially when the constriction is narrow. Despite this, the first eigenchannels are reasonably preserved for the “negative” case, even in the presence of scatterers, which is especially noticeable for the narrow constrictions, where two single eigenchannels are responsible for the first step to reach a transmission of two, plus some smaller contributions from the other eigenchannels. The degree of valley-degeneracy can be noted from the two rather closely spaced contour lines of the same colors shown in the top panels of Figure 11B,C, and is also noticeable in Figure 12B,C. The main reason for this comes from local pseudo-spin degeneracy being broken by the presence of nitrogen scattering potentials on random lattice sites. The transmission eigenvalues of the “normal” case in Figure 10C,E and in Figure 12B,C do not come close to unity. Again, this is due to the increased amount of nitrogen inside the constriction causing scattering in and mixing of the channels.

#### Shot Noise

Finally, we consider the shot noise, as the signal-to-noise ratio is an important characterization of quantum devices, and can give insights into the contributing channels in experiments. The Fano factor, *F* is defined as the ratio of actual noise to the full shot noise 2eI from a Poissonian current (*I*) in the device. It is given by,
(1)$$F=\frac{\sum_{n}\tau_{n}\left(1-\tau_{n}\right)}{\sum_{n}\tau_{n}}\;.$$

In the diffusive regime with a low number of low conducting channels, we expect F→1. For graphene in the ballistic limit, it has been predicted that F=1/3 for ideal graphene with a width much longer than the length of the conductor, W≪L, for energies near the Dirac point [37]. However, for disordered constrictions this is not likely to hold, as shown in simulations by Marconcini et al. [38]. Furthermore, in our case, we have W∼L.

The Fano factors for the “negative” and “normal” systems are seen in Figure 13 and Figure 14. From these, it is evident that there exists a high noise regime and a suppressed noise-to-signal regime for both pristine systems.

It is seen from Figure 13 and Figure 14 that only the “negative” case preserves the quantum regime well, where the Fano factor increases to ∼0.4. In contrast, for the “normal” case, the Fano factor is ∼0.7. The addition of nitrogen scatterers also moves the place where the transition between these regions happens, downwards in energy in both cases. This may be a result of the lowering of the average onsite potential in the presence of nitrogen atoms. From Figure 13, the distance from the charge neutrality point where the transition from high to low noise-to-signal occurs increases when the constriction becomes more narrow.

## 4. Discussion

The lensing shown in Figure 5A is originating from the Klein tunneling phenomenon described by Allain et al. [3]. But, the fact that it remains in the case of nitrogen scatterers shows that this phenomenon is somewhat robust to the local disturbances in the symmetry in sub-lattice sites in the graphene lattice for random doping. This is also a feature in the constriction calculations, where the valley-splitting seen in the contour lines in constrictions from each pseudo-spin are relatively close, except for Figure 14C. In this sense, the system properties are close to those of pristine graphene.

However, the transmission function for the constrictions shows a significant asymmetry between the electron and hole side of the Dirac point in the constriction. For clean constrictions, the quantization/step features only appear for energies starting at the constriction Dirac point going towards the Dirac point of the surroundings, corresponding to the electron (“normal”) and hole (“negative”) sides, respectively. For energies opposite to the quantization region, we note that the confinement is lost. This asymmetry is known from p−n junctions in graphene [39] and reflects the different confinement depending on the nature of the carrier.

A significant degradation of quantized conductance is seen in Figure 9 and Figure 10 when the nitrogen scatterers are introduced. This may also be because a narrower constriction collects waves from a smaller cross-section in the lead. Thus, the wave in the narrow constriction may be more coherent compared to the wide cross-section, where the waves have met a wider array of configurations of nitrogen scatterers, giving a higher degree of randomness in the phases of the waves that are confined to the constriction. This might be the reason why the first transmission step can still be observed at lower widths for both geometries in Figure 9D,F, but with a slightly more well-defined step in the case of a pristine junction.

The strength of the nitrogen scattering centers is, therefore, a crucial factor and it might be possible that steps can be taken to reduce it. For example, doping by potassium could possibly reduce the scattering. This has been shown to be experimentally feasible in graphene [40], and it has been demonstrated how scattering by potassium alone is low in carbon nanotubes, partly due to its position on top of carbon atoms, and partly due to its electron donation [21]. Modifying the charge carrier density has also been shown to be possible with graphene oxide films, where the carrier type can be chosen by annealing conditions [15,41]. It may be possible to use a combination of the C${}_{60}$-masking, possibly to obtain both types of doping, while also keeping the scattering low, because of the out-of-plane bond, as is the case with potassium.

The Fano factor from Figure 13 and Figure 14 furthermore shows that if this type of junction were to be used for sensing, the blue region is optimal for a low noise-to-signal ratio. These figures also show the different types of junctions should be operated with opposite majority charge carriers to achieve this.

Furthermore, the results demonstrate that the current path can be controlled in-plane through the doping, while keeping the graphene sheet planar. This could be combined with out-of-plane stacking into heterostructure devices [42], where one could imagine building guiding channels into the individual graphene sheets in a stack enabling control of the current pathways in three dimensions.

## 5. Conclusions

We have theoretically studied graphene systems with controlled, sharply defined patterns of different nitrogen concentrations which have recently been achieved experimentally [14]. We used Greens function-based large-scale transport methods using tight-binding (TB) electronic structure with DFT-based parameters. We benchmarked the transport properties of the TB model against full DFT for small, model systems. We considered how nitrogen dopant scattering centers affect current focusing effects p−n-junction systems, and the current confinement/guiding and quantization in constrictions formed by the different doping regions. In all cases, we found that the nitrogen dopant scattering plays a decisive role, but that the effects, such as current focusing and guiding, may still be observed in suitable energy ranges. Since the graphene is kept planar with this technique, this could also have applications in hetero-structure devices. 

## Figures and Tables

**Figure 1 nanomaterials-13-02556-f001:**
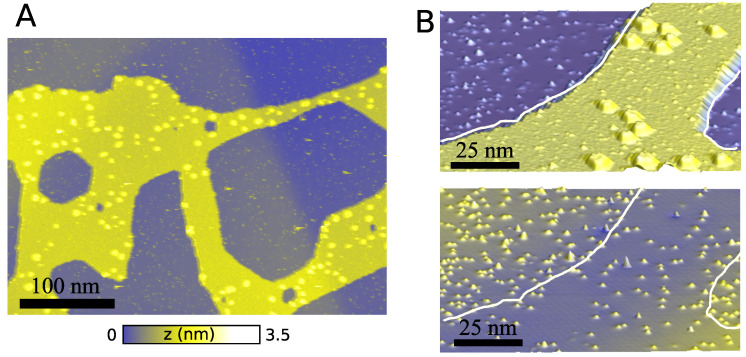
(**A**) STM image of HOPG with a C${}_{60}$ layer exposed to activated nitrogen (2 V, 5 pA). The yellow area corresponds to the C${}_{60}$ layer. The little spots on the surrounding graphene part correspond mainly to substitutional nitrogen. (**B**) (**Top**) STM image at a smaller scale showing a C${}_{60}$ layer on HOPG after nitrogen plasma exposure (2 V, 10 pA). (**Bottom**) STM image of the same area after removing the molecules with the STM tip (2 V, 50 pA). The white lines are a guide for the eye, indicating the limit of the C${}_{60}$ island.

**Figure 2 nanomaterials-13-02556-f002:**
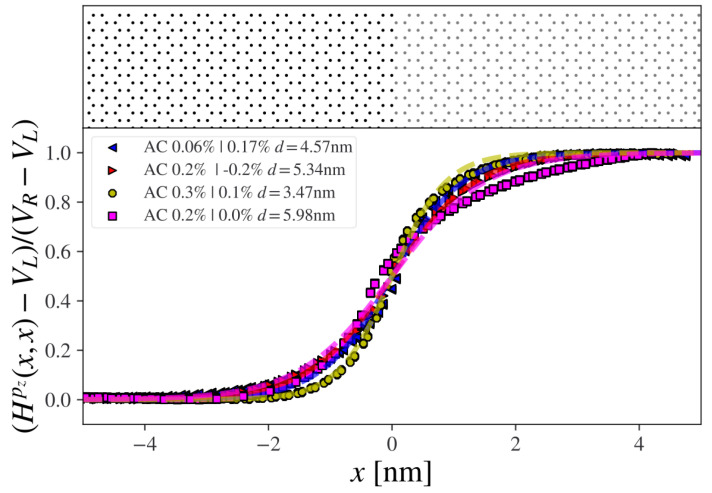
The $p_{z}$-onsite potential extracted from a DFT-VCA calculation for abrupt junctions (armchair direction) with different dopings. The junction width d=4b is found by the fitting function, given by $f\left(x\right)=A+B\tanh\left(\frac{x-c}{b}\right)$.

**Figure 3 nanomaterials-13-02556-f003:**
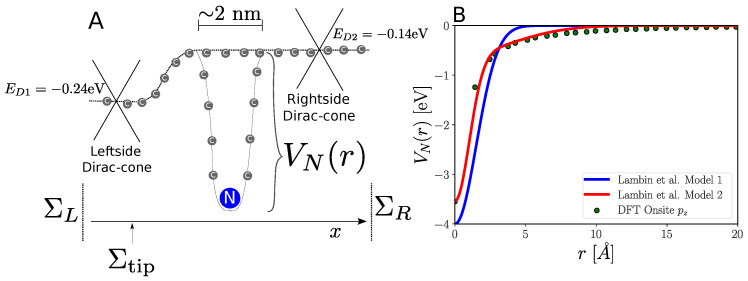
(**A**) Schematic of the tight-binding model consisting of a sum of a smooth junction-potential varying between the left-side Dirac point ($E_{D1}$) and right-side Dirac point ($E_{D2}$) (cf. Figure 2), short-ranged nitrogen potential $V_{N}\left(r\right)$, and self-energies Σ accounting for left/right electrodes. (**B**) Comparing fits of distance dependent $p_{z}$-onsite nitrogen potentials based on a single Gaussian (Model 1) and two Gaussians (Model 2) to DFT calculation (points).

**Figure 4 nanomaterials-13-02556-f004:**
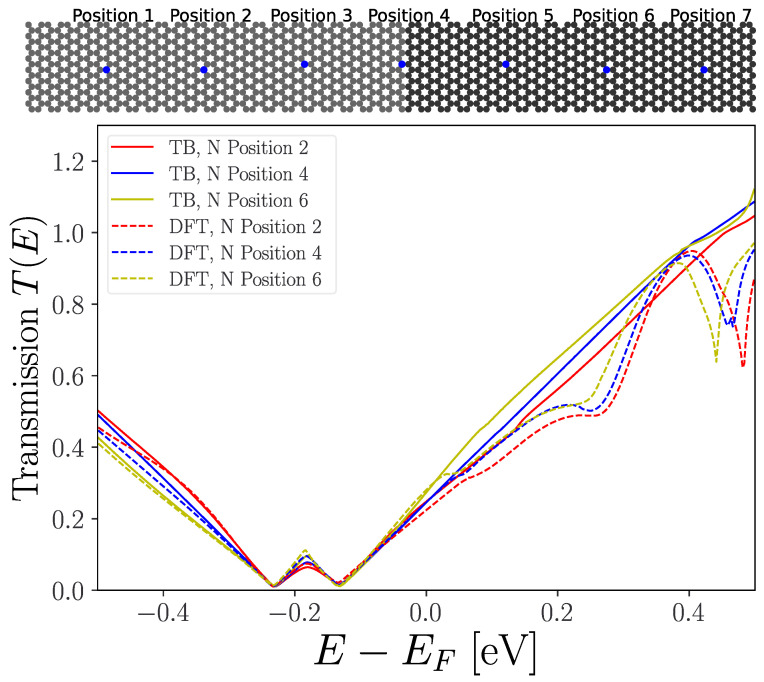
Benchmark calculations: The tight-binding (TB) compared to full DFT electron transmission calculations for different nitrogen positions with respect to transport across the planar junction. Parameters for the junction are n=0.17% and $n^{\prime }=0.06\%$, corresponding to the percentage of nitrogen atoms among the carbon atoms in the experiment [14]. Periodic boundary conditions are applied in the transverse *y*-direction. The two transmission minima correspond to the Dirac points for the two doping levels.

**Figure 5 nanomaterials-13-02556-f005:**
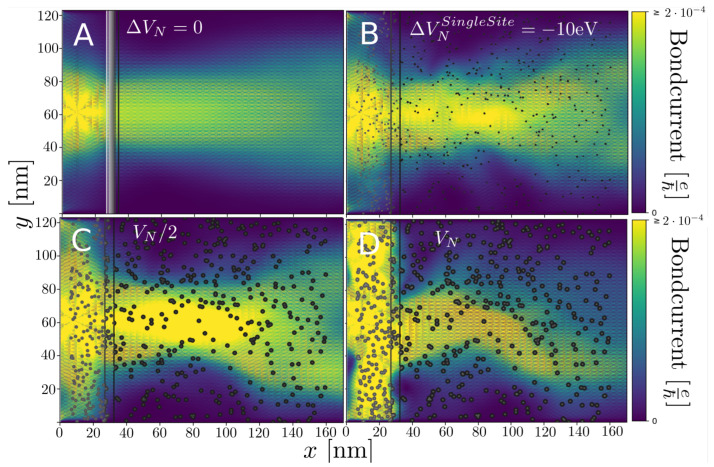
Spectral bondcurrents injected by a point source (STM tip) at a specific energy (tip voltage) tuned for good wave focusing comparing different N-potential parameters. (**A**) Pristine, straight junction system without nitrogen scattering. (**B**) N-only onsite shift TB model fitted to DFT/T-matrix [22]. (**C**,**D**) Distance-dependent TB model (cf. Figure 2, Model 2 [19]) with 50% reduction in $V_{N}$ (**C**), and without reduction (**D**).

**Figure 6 nanomaterials-13-02556-f006:**
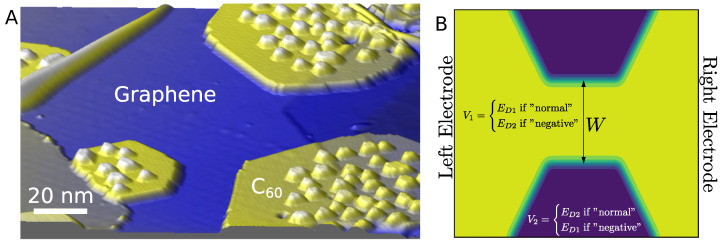
(**A**) Experimental STM image showing several constrictions formed in the C${}_{60}$-overlayers (corrugated regions) on graphene. The width of the narrowest constriction is around 20 nm. We denote the case where the constrictions are not covered (high doped; the “normal” case), while the reverse (low doped constriction) is the “negative” case. (**B**) Simulation setup of a single constriction. Semi-infinite left/right electrodes are applied, and complex absorbing potentials (CAPs) are used in the transverse directions. The two Dirac points are indicated together with the variable constriction width, *W*.

**Figure 7 nanomaterials-13-02556-f007:**
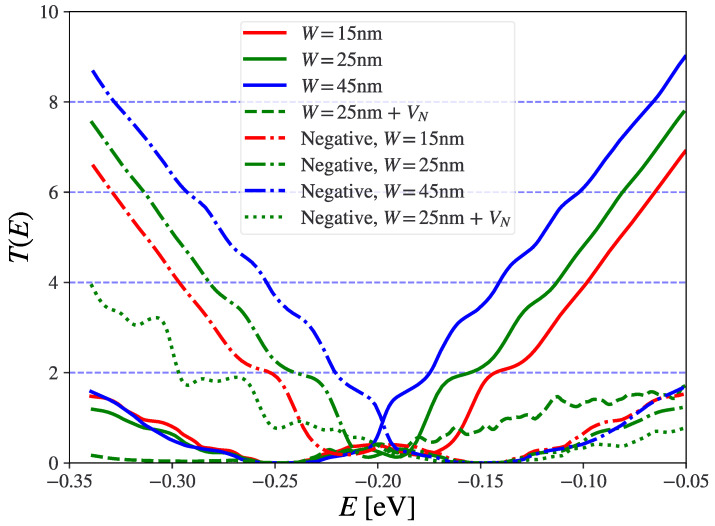
Transmission vs. energy for “normal” and “negative” constrictions of various widths (*W*), and with/without nitrogen scattering potential for W=25 nm ($+V_{N}$). The ”normal” setup has the constriction $n^{\prime }$-doped with 0.17% N corresponding to $∼E_{D}=-0.24$ eV surrounded by the masked *n*-doped region with 0.06% N with $∼E_{D}=-0.14$ eV, and reversed for ”negative”.

**Figure 8 nanomaterials-13-02556-f008:**
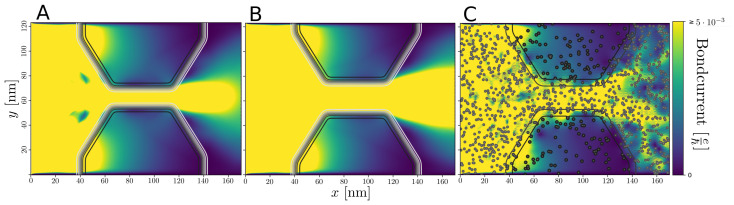
Spectral bondcurrents injected in the left wide $n^{\prime }$-region with $E_{D1}=-0.24$ eV at an energy E=−0.155 eV, while it is surrounded by a less doped *n*-region with $E_{D2}=-0.14$ eV corresponding to the “normal” constriction. (**A**) Narrow constriction W=15 nm. (**B**) Wider constriction, W=25 nm. (**C**) Wide constriction, W=25 nm, with nitrogen potentials included.

**Figure 9 nanomaterials-13-02556-f009:**
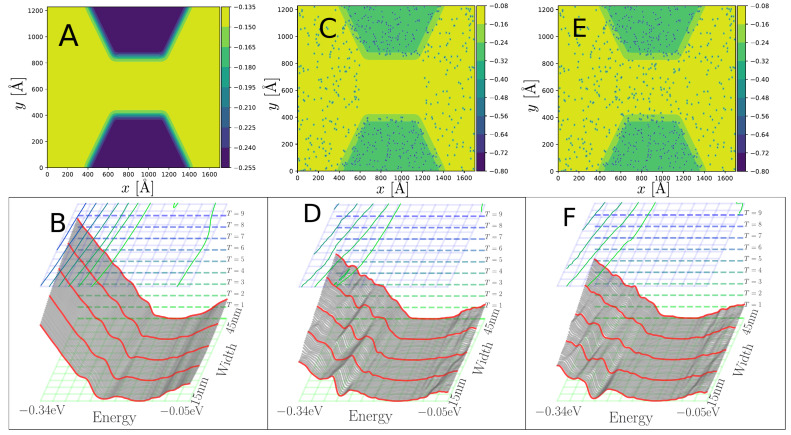
Transmission function as a function of energy and width, T(E,W), for the “negative” type constriction. (**A**) Pristine constriction, and (**B**) corresponding transmission. (**C**) Constriction without N scattering in the narrowest part, and (**D**) corresponding transmission. (**E**) Highly doped constriction, and (**F**) corresponding transmission. The contours in the top part of (**B**,**D**,**F**) correspond to the integers T=n, and lines seen in the background of (**B**,**D**,**F**) are color-coded accordingly.

**Figure 10 nanomaterials-13-02556-f010:**
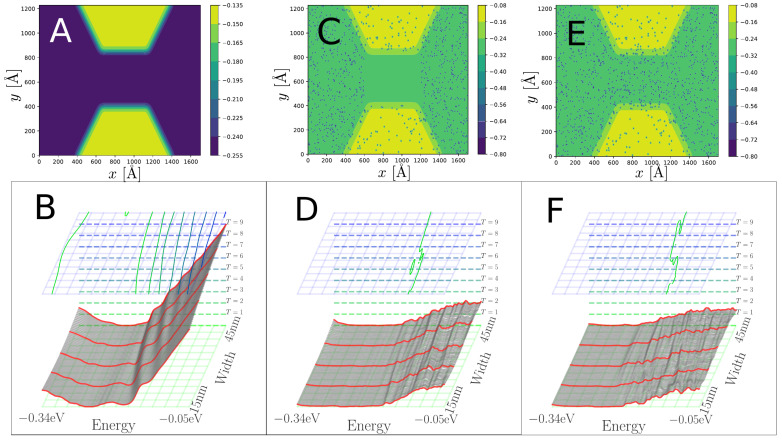
Transmission function as a function of energy and width, T(E,W), for the “normal” type constriction. (**A**) Pristine constriction, and (**B**) corresponding transmission. (**C**) Constriction without nitrogen in the narrowest part, and (**D**) corresponding transmission. (**E**) Highly doped constriction, and (**F**) corresponding transmission. The contours in the top part of (**B**,**D**,**F**) correspond to the integers T=n, and lines seen in the background of (**B**,**D**,**F**) are color-coded accordingly.

**Figure 11 nanomaterials-13-02556-f011:**
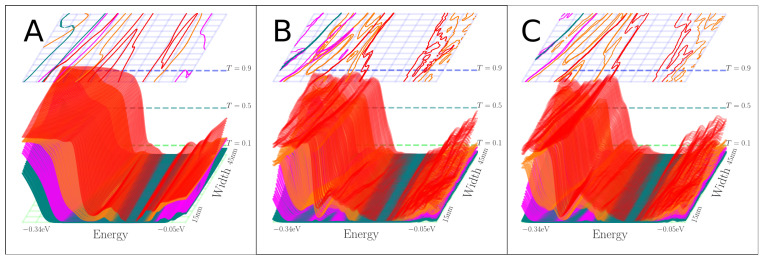
Transmission eigenvalues as function of energy *E* and constriction width *W* for the “negative” case of Figure 7. The colors (red, green, blue, dark blue) reflect the approximately valley-degenerate channels. (**A**) Corresponds to Figure 9A. (**B**) Corresponds to Figure 9C. (**C**) Corresponds to Figure 9E. Each time the surface intersects one of the horizontal planes defined by the dashed lines in the background, a line with the same color is drawn in the top panel.

**Figure 12 nanomaterials-13-02556-f012:**
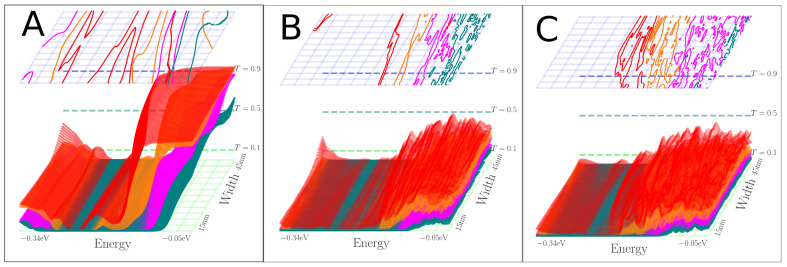
Transmission eigenvalues as function of energy *E* and constriction width *W* for the “normal” case of Figure 7. The colors (red, green, blue, dark blue) reflect the approximately valley-degenerate channels. (**A**) Corresponds to Figure 10A. (**B**) Corresponds to Figure 10C. (**C**) Corresponds to Figure 10E. Same plotting method as in Figure 11.

**Figure 13 nanomaterials-13-02556-f013:**
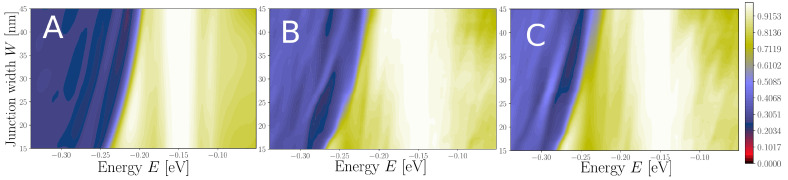
(**A**–**C**) Fano factor of the device setup illustrated in Figure 9A,C,E.

**Figure 14 nanomaterials-13-02556-f014:**
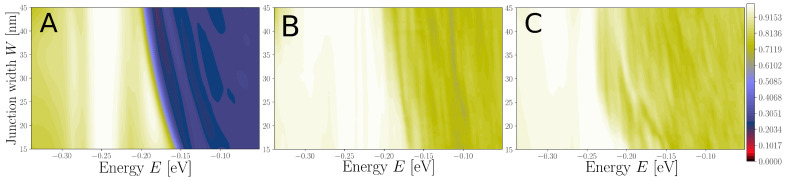
(**A**–**C**) Fano factor of the device setup illustrated in Figure 10A,C,E.

## Data Availability

Not applicable.

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
