# Peer review of "Quantum Transport in Large-Scale Patterned Nitrogen-Doped Graphene"

_nanomaterials, 2023, doi:10.3390/nano13182556_

Round 1

Reviewer 1 Report

The paper presents new and valuable results on theoretical investigation of graphene and doped graphene systems with controlled level of nitrogen dopant. The approach used Greens function based large-scale transport methods using tight-binding (TB) electronic structure with DFT-based parameters.

It was supposed that nitrogen dopant scattering centers affect current focusing effects in p − n-junction systems, and the current confinement/guiding and quantization in constrictions formed by the different doping regions.

As to reviewers comments,

     I would note that atomic composition, interconnection of carbon, oxygen and N–dopant atoms is critical for a carbon-based device performance. Oxygen atoms might be (though not necessary) additionally allowed within the model as if the model uses some experimental results from the earlier studies. Particularly, graphene oxide might be present along with pure grapheme as a result of the synthetic procedure. Perhaps, it would be a helpful for a reader if the introduction section would contain a few lines of text devoted to some research results on atomic composition, for example, XPS results of graphene-like materials or small conjugated molecular films materials. Below are some refs on the subject, which I found as examples.

1. Komolov A.S., et.al. Materials and design, 2017, 113, 319 DOI: http://dx.doi.org/10.1016/j.matdes.2016.10.023

2. Komolov, A.S., et al., 2019, Journal of Electron Spectroscopy and Related Phenomena. 235, 40-45. http://dx.doi.org/10.1016/j.elspec.2019.07.001

 I recommend this manuscript for publication after minor revision.  

Regards, Reviewer

---------

----------

Author Response

Reviewer 1:

I would note that atomic composition, interconnection of carbon, oxygen and N–dopant atoms is critical for a carbon-based device performance. Oxygen atoms might be (though not necessary) additionally allowed within the model as if the model uses some experimental results from the earlier studies. Particularly, graphene oxide might be present along with pure grapheme as a result of the synthetic procedure. Perhaps, it would be a helpful for a reader if the introduction section would contain a few lines of text devoted to some research results on atomic composition, for example, XPS results of graphene-like materials or small conjugated molecular films materials. Below are some refs on the subject, which I found as examples.

We thank the reviewer for the suggestion on the inclusion of oxygen in the device. The graphene oxide is not relevant, since the synthesis is done in UHV by annealing a SiC wafer, there is no exposure to oxygen during the sample preparation process. The experimental setup is using an ultra-high vacuum setup, which makes it very unlikely that oxygen will be present in the chamber. On the other hand, in an actual device that could be exposed to the atmosphere it is definitely a concern, but encapsulation methods could be used to get around this.

The following sentences in red has been added in the discussion:

“…. and it has been demonstrated how scattering by potassium alone is low in carbon nanotubes, partly due to its position on top of carbon atoms, and partly due to its electron donation[15]. Modifying the charge carrier density has also been shown possible with graphene oxide films, where the carrier type can be chosen by annealing conditions[33, 34]. It may be possible to use a combination of the C60-masking to possibly obtain both types of doping, while also keeping the scattering low, because of the out-of-plane bond, as is the case with potassium.”

and

“This gives a potential profile that corresponds to the shape of the covering layer when the C$_{60}$ is removed from the surface by sweeping the STM tip[Bouatou2022]. Other interesting methods are also available for doping graphene, such as annealing of graphene oxide, but lacks that nano-scale precision of the method developed in ref. [10]. “

with a new reference to the first article suggested by the reviewer, and another article on the subject.

Reviewer 2 Report

The manuscript cannot be accepted in its present form as there are several issues associated with it. Hence, a major revision is required before acceptance. The authors must address the following concerns properly:

1) The abstract section should not include the citation of the authors' previous work. Moreover, the last few lines should also be focused on the implications of the study.

2) The introduction section should clearly present the state-of-the-art. There should be a proper discussion on the nitrogen (N) doping of graphene/GNRs as there are several research reports on N-doping of graphene. The introduction section should be streamlined according to this. Moreover, in-plane conductivity and optoelectronic properties should also be discussed briefly by going through some important articles such as https://pubs.acs.org/doi/abs/10.1021/acsaelm.1c01350; https://www.science.org/doi/full/10.1126/science.aaz2570; https://iopscience.iop.org/article/10.1088/1361-6528/ac8e0e/meta; https://pubs.acs.org/doi/abs/10.1021/nn5049188

3) What do you mean by n-n' junction (in the abstract)? Also, there is no discussion of it in the manuscript

4) There is confusion about sample preparation. Is it HOPG or graphene film? What about using a different mask other than C60?

5) Are you considering it as a graphene film or nanoribbon (GNR)?

6) The calculations have been carried out for electron transport whereas N-doping turns graphene p-type. Is there any difference? Are DFT and other calculations performed at room temperature?

7) What about the effect of defects created after N doping? If the sample is considered to be GNR, then the ballistic transport should be taken into account. Moreover, clear schematics of the energy band diagrams should be presented showing variations in Fermi levels.

Author Response

Reviewer 2:

We thank the reviewer for giving concrete points of critique to be improved upon.

1) The abstract section should not include the citation of the authors' previous work. Moreover, the last few lines should also be focused on the implications of the study.

We have modified the abstract to account for the reviewer’s suggestion. The citation has been removed.

2) The introduction section should clearly present the state-of-the-art. There should be a proper discussion on the nitrogen (N) doping of graphene/GNRs as there are several research reports on N-doping of graphene. The introduction section should be streamlined according to this. Moreover, in-plane conductivity and optoelectronic properties should also be discussed briefly by going through some important articles such as https://pubs.acs.org/doi/abs/10.1021/acsaelm.1c01350; https://www.science.org/doi/full/10.1126/science.aaz2570; https://iopscience.iop.org/article/10.1088/1361-6528/ac8e0e/meta; https://pubs.acs.org/doi/abs/10.1021/nn5049188

We agree with the reviewer that more discussion on nitrogen with graphene can improve the paper. However, there are a number of review papers on this topic already. Therefore, we have added extra references to these.

A line in the introduction with references to reviews.

“Nitrogen doped graphene have focused a wide interest for fundamental and applied research in various fields as outlined by several review articles [1-4]”

with citations:

[1] H. Wang, T. Maiyalagan, X. Wang, Review on recent progress in nitrogen-doped graphene: Synthesis, characterization, and its potential applications, Acs Catal. 2012, 2, 781.

[2] X.-K. Kong, C.-L. Chen, and Q.-W. Chen, Doped graphene for metal-free catalysis, Chem. Soc. Rev. 43, 2841 (2014).

[3] R. Yadav, C. K. Dixit, Synthesis, characterization and prospective applications of nitrogen-doped graphene: A short review, J. Sci. Adv. Mater. Devices 2017, 2, 141.

[4] F. Joucken, L. Henrard, J. Lagoute, Electronic properties of chemically doped graphene, Phys. Rev. Mater. 2019, 3, 110301.

However, we do not address optoelectronic/photodetection properties at all in our work which is the focus in the following suggested references:

https://pubs.acs.org/doi/abs/10.1021/acsaelm.1c01350

https://iopscience.iop.org/article/10.1088/1361-6528/ac8e0e/meta

The reference https://www.science.org/doi/full/10.1126/science.aaz2570 is about co-axial nanotubes and https://pubs.acs.org/doi/full/10.1021/nn5049188 on bilayer graphene.

We feel these suggested references are less relevant to our work to merit citation.

3) What do you mean by n-n' junction (in the abstract)? Also, there is no discussion of it in the manuscript

Added explanation of carrier types in p-n and n-n’ junctions:

Changed:

p-n or n-n’

To:

“...form of p − n (holes are carriers on one side, electrons the other) or n − n′ (different electron 27
concentrations)[4] junctions to enable electron….”

4) There is confusion about sample preparation. Is it HOPG or graphene film? What about using a different mask other than C60?

It is HOPG graphene that is true, and the connection from HOPG to the graphene monolayer we model is not made clear in the article. Exfoliation using pick-up techniques can be used to move a monolayer flake of graphene into a device.

The following sentence (in red) has been added:

“...we show Scanning Tunnelling Microscopy (STM) images of a C60 overlayer mask on graphene. Here the outermost graphene layer of a slab of highly oriented pyrolytic graphite (HOPG) is seen”

The referee is right that other molecules might be used as a mask, but we are not aware of work employing other molecules as the mask in a similar setup with nitrogen doping. We have added the efficiency of the process is 60%:

We have changed the text to make this clearer:

When doping graphene in the lab, the different concentrations of nitrogen are implanted by first covering parts of the graphene with a C60 mask, then bombarding the surface with nitrogen, yielding a 60% reduction in nitrogen implantation under the mask[10].

5) Are you considering it as a graphene film or nanoribbon (GNR)?

Simulations are done such that it is a graphene film with open boundary conditions in order to avoid spurious edge effects. CAPs are used for this purpose. We believe this mimics the experiments on large graphene samples better than a ribbon-approximation.

We have changed the text to make this clear:
In particular, complex absorbing potentials (CAPs) are used to absorb current and avoid spurious edge effects which would appear in a (finite) ribbon approximation”

6) The calculations have been carried out for electron transport whereas N-doping turns graphene p-type. Is there any difference? Are DFT and other calculations performed at room temperature?

We stress that the implanted N-atoms donates electrons to the graphene sheet, n-doping it.  The calculation includes both electrons and holes, but in a single-particle picture. So it should not matter. DFT calculations are performed at room temperature which is primarily useful for convergence of the electronic density.

The following sentence in red has been added:

“We used the PBE exchange-correlation functional together with 20 k-points in the transverse (y-) direction and an electron smearing of 0.025eV. “

7) What about the effect of defects created after N doping? If the sample is considered to be GNR, then the ballistic transport should be taken into account. Moreover, clear schematics of the energy band diagrams should be presented showing variations in Fermi levels.

This is an interesting question, to which we at the moment have no good answer to other than to say we assume the bonds are not broken in the model. Disorder of this type is another good idea to investigate, but we have not included this in our modeling.

The transport theory we use explicitly assumes the electron is traveling ballistically/elastically throughout the modeling region. The currents we calculate are meant to be interpreted as currents induced by a small voltage.

The Fermi-level is the same everywhere, as we only consider equilibrium phenomena. The position of the Fermi-level relative to the Dirac point is in turn globally tunable by external input, such as a back-gate.

To clarify this we have added:

Fig.3A does show the variation of the Dirac point throughout the graphene sheet we model, corresponding to measurements in ref. Bouatou et al

To make this more clear, we have made changes to the text:
“We split the contribution of the nitrogen dopants into a smooth background potential, as modeled in the spirit of VCA corresponding to the spacial variation in the Dirac point measured by STM spectroscopy[ 10 ], and a short-ranged scattering potential centered at the nitrogen atoms. This is schematically shown in Fig. 3A.”

and in the caption of Fig. 3:

”Schematic of the tight-binding model consisting of a sum of a smooth junction-potential
varying between the left side Dirac point (ED1) and right side Dirac point (ED2) (cf. Fig. 2), short-ranged
nitrogen potential VN(r) and self-energies Σ accounting for left/right electrodes”

Reviewer 3 Report

In the manuscript, the authors present the results of DFT calculations of nitrogen doped graphene systems, which relate directly to experimental systems in which the charge transport properties (to what extent ballistic quantum transport and conductance quantization is preserved) were studied. Since the work contributes new knowledge about the mechanisms of charge transport in graphene systems, I recommend it for publication in Nanomaterials.

The authors are asked to correct Figure 5. In the legend to the current map on the right sight of the plot, the maximum value is marked as 2,000e-C. What does 'C' mean?.

Author Response

Reviewer 3:

We have made the following improvements to our figures:

Fig. 5 has been cropped correctly (reading now 2.0e-04 instead of 2.0e-C). Thank you for noticing.

Fig. 8 has furthermore had a bondcurrent label added to it.

Fig. 13 and Fig. 14 have been updated to have better resolution and without pdf-rendering artifacts.

Reviewer 4 Report

Graphene holds remarkable electronic transport properties, quantum conductors like graphene nanoribbons reveal astonishing characteristics. Their few electronic states coupled with minimal electron-phonon scattering stem from the inherent symmetry of quantum states. These properties inspire concepts of electron-based optics and devices utilizing Dirac fermion behavior.

To harness this potential, precise interfaces at the Fermi wavelength scale are crucial, enabling electron current focus and innovation like Veselago lenses and novel transistors. Nitrogen doping emerges as a natural choice, introducing carrier scattering with atomic-scale precision. The study employs transport calculations rooted in graphene's N-doping interfaces, echoing real experiments. It begins with comparing tight-binding models and density functional theory for small systems. Subsequently, large-scale transport calculations probe electron currents and conductance quantization in junctions and constrictions, unveiling the role of N-scattering and disorder.

It is recommended to simplify language (replace complex phrases with simpler equivalents for easier comprehension), enhance flow (organize sentences for smoother transitions between concepts), condense Details (focus on core points) and most importantly, highlight significance (emphasize the importance of the study's approach and findings).

HOPG, LCAO, PBE – any abbreviation used should be mentioned beforehand at its first text occurrence. Top of Form

The correct electron structure of carbon is 1s2 2s2 2px1 2py1 , hence the modelling on line 86 needs attention (“carbon atom is modelled using a single pz-orbital”), since the TB model only account for the pz-only model (lines 99-100)

How were the C60 molecules placed and displaced from graphene? Have the authors analyzed the impact of N-impact on fullerene as well?

In Figure 4, the position of N atoms is almost impossible to read, so the positions 1-7 should be redrawn using more easily distinguishable colors.

The work in general seems more appropriate to computational journals rather than Nanomaterials, since it revolves around calculations rather than practical importance/applications of these N-doped graphene by C60 masking. Have the authors considered any other marking other than C60? Please consider previously published (and cited) references like [6], [9], [12], [14], [17], [19], [28] etc.

As a side-note, 7 out of 32 publications refer to previous publications from the group (self-citations), which is quite high (~ 22%).

The text should also benefit from data presented using tables; this would make the draft easier to read, see for instance the case of Figs. 13 and 14.

The draft is not properly structured; the Discussion section has literally three paragraphs, when it should have contained a lot more data and graph analysis; this could be done by moving and rewriting parts of Section 3 and adapting them for Section 4.

The text is very “dry” from materials’ scientist point of view, and the significance of the content is not apparent. What are the main findings, and how would they impact graphene technology as we know it today? What direct implications and potential applications would the N-doping by masking have? The authors should address this in the manuscript, throughout the draft.

To conclude, the investigation focused on the theoretical analysis of experimentally achieved controlled nitrogen concentration patterns in graphene. Leveraging Greens function-based large-scale transport methods with tight-binding (TB) electronic structure, underpinned by DFT-based parameters, the authors established a robust TB model, validated against DFT for smaller systems.

The draft would only need minor editing during proofreading, speaking strictly from the English viewpoint.

Author Response

Reviewer 4:

We thank the reviewer for pointing to places of improvement. We have addressed:

1) It is recommended to simplify language (replace complex phrases with simpler equivalents for easier comprehension), enhance flow (organize sentences for smoother transitions between concepts), condense Details (focus on core points) and most importantly, highlight significance (emphasize the importance of the study's approach and findings).

HOPG, LCAO, PBE – any abbreviation used should be mentioned beforehand at its first text occurrence.

Changes: We removed “LCAO” from the sentence. It was a superfluous word.

Added the following sentences in red:

“We used the Perdew-Burke-Ernzerhof (PBE) exchange-correlation functional together with…”

“...overlayer mask on graphene. Here the outermost graphene layer of a slab of highly oriented pyrolytic graphite (HOPG) is seen.”

We have added the following to the abstract:

“The implications for device design is that a p-n junction with nitrogen corrugation should still be viable for current focusing. Furthermore, a guiding channel with less nitrogen in the conducting canal preserves more features of quantized conductance and therefore its low-noise regime”

2) The correct electron structure of carbon is 1s2 2s2 2px1 2py1 , hence the modelling on line 86 needs attention (“carbon atom is modeled using a single pz-orbital”), since the TB model only account for the pz-only model (lines 99-100)

The planar structure of graphene means that any even and any odd function (with zero located at the graphene plane) has no coupling, effectively giving a frozen subsystem consisting of the orbitals 1s 2s 2px 2py (all even functions) while the pz orbital is the only orbital changing sign through the z=0 plane. Furthermore the (1s 2s 2px 2py) part is not close in energy to the Fermi-level, only the (pz) part, which is why graphene can be modeled using effectively a single orbital. The rest plays no role when the planar symmetry is not broken. Nitrogen dopants does not do this either.

The following sentence in red has been added:

“For in-plane doping it is a natural choice to dope graphene by substituting carbon with nitrogen due to similar atomic radii and sp$^2$-bonding with similar bond-lengths, leading to low lattice deformation and planarity. This furthermore gives rise to a pi-system of electrons at the Fermi-level, well-described by a widely used $p_z$-model. “

3) How were the C60 molecules placed and displaced from graphene? Have the authors analyzed the impact of N-impact on fullerene as well?

The following sentence in red has been added:

“This gives a potential profile that corresponds to the shape of the covering layer when the C60 is removed from the surface by sweeping the STM tip”

4) As a side-note, 7 out of 32 publications refer to previous publications from the group (self-citations), which is quite high (~ 22%).

We have added more references to improve on the reviewing part of our manuscript.

5) The text should also benefit from data presented using tables; this would make the draft easier to read, see for instance the case of Figs. 13 and 14.

Thank you for the suggestion, but we do not see how these figures can be put into tables for a clearer presentation. In our opinion it is the colored regions which carry the message here.

6)

The draft is not properly structured; the Discussion section has literally three paragraphs, when it should have contained a lot more data and graph analysis; this could be done by moving and rewriting parts of Section 3 and adapting them for Section 4.

Thank you for the suggestions.

We have made the following changes:

A paragraph from the results section on the possible causes of the results has been moved to the discussion section. Another paragraph has been added to discuss the meaning of the Fano factor plots. Lastly, a short paragraph about the applications to hetero-structures have been added.

We do, however, respectfully disagree on the content of the discussion section; We feel that it should not present new results. We further note that the 3 other reviewers did not make a notice of this.

Paragraph moved to discussion:

“A significant degradation of quantized conductance is seen in Fig. 9 and Fig. 10 when the nitrogen scatterers are introduced. This may also be because a narrower constriction collects waves from a smaller cross-section in the lead. Thus, the wave in the narrow constriction may be more coherent compared to the wide cross-section, where the waves have met a wider array of configurations of nitrogen scatterers, giving a higher degree of randomness in the phases of the waves that are confined to the constriction. This might be the reason why the first transmission step can still be observed at lower widths for both geometries in Fig. 9D and Fig. 9F, but with a slightly more well-defined step in the case of a pristine junction.”

Paragraph added to discussion:

“The Fano factor from Fig. 13 and Fig. 14 furthermore shows that if this type of junction where to be used for sensing, the blue region is optimal for a low noise-to-signal ratio. These figures also shows the different types of junctions should be operated with opposite majority charge carriers to achieve this.”

Added to discussion:

“Furthermore, the results demonstrate the current path can be controlled in-plane through the doping, while keeping the graphene sheet planar. This could be combined with out-of-plane stacking into heterostructure devices[ 40 ], where one could imagine building guiding channels into a graphene sheet and stacking them on top of each other.“

Other Corrections

Fig. 2 cation:

  1. B) The junction width d = 4b is found by fitting function given by f (x) =A + B tanh( (x−c)/b)

Round 2

Reviewer 2 Report

The revised manuscript is still not publishable as the authors have not addressed the queries properly.

1) Mere writing a couple of lines regarding the N-doping of graphene by citing a few review articles does not change anything as the N-doping of graphene is the main component of the manuscript.

2) N-doping of graphene can also turn graphene into a p-type for example, https://pubs.acs.org/doi/10.1021/acsanm.8b02237. Hence, it is very much important to know which type of graphene has been considered after N-doping.

3) The discussion about the Fermi level positioning becomes important.

4) I was wondering as there are no experimental measurements of quantum transport properties of the N-doped graphene as the present study is not purely theoretical.

5) The theoretical results are loosely correlated with the experimental results.

6) According to the authors, the top graphene layer became n-type after N-doping, and there exists a variation in charge carrier concentration depending upon the exposed (high doping) and unexposed (low doping due to the C60 mask) area but in any case, it is n-type (as per authors' explanation), then what do you mean by p-n junction (holes are carriers on one side, electrons the other)? I think here comes the experimental verification to establish such a phenomenon.

Author Response

We thank the reviewer for hers/his comments which we believe have made our text clearer.
We agree in part and have made changes accordingly to clarify the remaining points.

Reviewer 4 Report

Most of the issues have been addressed. 

Mostly fine, small errors should be corrected during proofreading. 

Author Response

We thank the reviewer for the comments.